# Goose Hepatic IGFBP2 Is Regulated by Nutritional Status and Participates in Energy Metabolism Mainly through the Cytokine−Cytokine Receptor Pathway

**DOI:** 10.3390/ani13142336

**Published:** 2023-07-17

**Authors:** Fangbo Li, Ya Xing, Jinqi Zhang, Ji’an Mu, Jing Ge, Minmeng Zhao, Long Liu, Daoqing Gong, Tuoyu Geng

**Affiliations:** 1College of Animal Science and Technology, Yangzhou University, Yangzhou 225009, China; qq971409175@outlook.com (F.L.); xingya325@163.com (Y.X.); zjq397495452@outlook.com (J.Z.); m18326768696@outlook.com (J.M.); gejing@yzu.edu.cn (J.G.); zhaominmeng123@163.com (M.Z.); liujiaolong688@sina.com (L.L.); 2Joint International Research Laboratory of Agriculture and Agri-Product Safety of the Ministry of Education of China, Yangzhou University, Yangzhou 225009, China

**Keywords:** IGFBP2, goose, liver, cytokine−cytokine receptor pathway, energy metabolism

## Abstract

**Simple Summary:**

Nutritional shortages or oversupply, such as fasting, restricted feeding, or overfeeding, often occur during animal production. Changes in nutritional status significantly affect animal health and production performance. Although the effect of nutritional changes on the expression of insulin-like growth factor-binding protein 2 (IGFBP2) in the liver is well established, the specific role of IGFBP2 in the response of the goose liver to these changes remains unclear. Further investigations are required to fully understand the involvement of IGFBP2 in this process. To address this, we used two types of animal models: the fasting/refeeding and overfeeding models of geese for in vivo study, and cultured goose primary hepatocytes for in vitro study. Data indicate that *IGFBP2* expression in the goose liver was induced by fasting, but inhibited by refeeding and overfeeding. *IGFBP2* overexpression in goose primary hepatocytes mainly inhibited the expression of the genes in the cytokine–cytokine receptor pathway, which was partly validated in in vitro models. These findings suggest that *IGFBP2* mediates the response of the goose liver to changes in nutritional status, mainly through the cytokine–cytokine receptor pathway.

**Abstract:**

Changes in the nutritional status of animals significantly affect their health and production performance. However, it is unclear whether insulin-like growth factor-binding protein 2 (IGFBP2) mediates these effects. This study aimed to investigate the impact of changes in nutritional and energy statuses on hepatic IGFBP2 expression and the mechanism through which IGFBP2 plays a mediating role. Therefore, the expression of *IGFBP2* was first determined in the livers of fasting/refeeding and overfeeding geese. The data showed that overfeeding inhibited *IGFBP2* expression in the liver compared with the control (normal feeding) group, whereas the expression of *IGFBP2* in the liver was induced by fasting. Interestingly, the data indicated that insulin inhibited the expression of *IGFBP2* in goose primary hepatocytes, suggesting that the changes in *IGFBP2* expression in the liver in the abovementioned models may be partially attributed to the blood insulin levels. Furthermore, transcriptome sequencing analysis showed that the overexpression of *IGFBP2* in geese primary hepatocytes significantly altered the expression of 337 genes (including 111 up-regulated and 226 down-regulated genes), and these differentially expressed genes were mainly enriched in cytokine–cytokine receptor, immune, and lipid metabolism-related pathways. We selected the most significant pathway, the cytokine–cytokine receptor pathway, and found that the relationship between the expression of these genes and *IGFBP2* in goose liver was in line with the findings from the *IGFBP2* overexpression assay, i.e., the decreased expression of *IGFBP2* was accompanied by the increased expression of *LOC106041919*, *CCL20*, *LOC106042256*, *LOC106041041*, and *IL22RA1* in the overfed versus normally fed geese, and the increased expression of *IGFBP2* was accompanied by the decreased expression of these genes in fasting versus normally fed geese, and refeeding prevented or attenuated the effects of fasting. The association between the expression of these genes and *IGFBP2* was verified by *IGFBP2*-siRNA treatment of goose primary hepatocytes, in which IGFBP2 expression was induced by low serum concentrations. In conclusion, this study suggests that IGFBP2 mediates the biological effects induced by changes in nutritional or energy levels, mainly through the cytokine−cytokine receptor pathway.

## 1. Introduction

During animal production, the nutritional status changes over time, affecting animal performance and health. For example, in piglets, restriction of the feed intake can decrease the weight, cross-sectional area, and glycogen content of the muscles [1]. In chickens, fasting slows the growth and development of the liver, pancreas, and spleen [2]. In contrast, overfeeding can increase the breast muscle thickness and body weight of chickens [3].

Fasting and refeeding, two completely different nutritional statuses, are often used to investigate how changes in nutritional status influence animal metabolism and physiology. The liver is an important organ in nutrient metabolism and plays an important role in nutrient conversion and distribution. Previous studies have indicated that fasting and refeeding can influence metabolite concentrations, enzyme activity, and mitochondrial respiration in chicken liver [4]. It has also been reported that fasting not only alters intrahepatic thyroid hormone levels (increased thyroxine or T4 levels and decreased triiodothyronine or T3 levels, which parallels changes in the plasma) [5], but also alters the phosphorylation level of the insulin receptor tyrosine kinase (decreased in the fasted state and increased in the refeeding state compared with the *ad libitum* fed state) in the chicken liver [6]. These findings indicate that changes in nutritional status may strongly affect metabolism and physiology in poultry liver, and fasting/refeeding is a good model for investigating the effects of changes in nutritional status, which has been demonstrated by many studies.

Overfeeding and high-sugar high-fat diet feeding are other models commonly used to address the effects of nutritional surplus on animal metabolism and physiology. Goose fatty liver (or *foie gras*) can be induced by overfeeding. Compared with *ad libitum* feeding, overfeeding in geese can lead to the up-regulation of genes related to the mitochondria, fatty acid desaturases 1/2, and adiponectin receptors 1/2. However, the expression of key inflammation-related genes *TNFα* and *C3* is suppressed, a phenomenon that differs from what is seen in mouse fatty liver [7]. Moreover, transcriptome analysis on goose fatty liver versus normal liver revealed that transcriptome is characterized by changes in the expression of genes related to metabolic pathways in the early stage of overfeeding, while transcriptome is characterized by changes in the expression of genes related to the cell growth and death pathway and the immune disease pathway in addition to the metabolic pathways in the late stage of overfeeding [8].

Therefore, IGFBP2 may mediate the biological effects of changes in the nutritional status of animals. IGFBP2 is a member of the IGFBP family, which includes IGFBP1-7. Based on the current annotation of reference sequences deposited in GenBank, human IGFBP2 has three isoforms that are 159, 181, and 325 amino acids long. All of the isoforms contain a “thyroglobulin type-1 repeat” region and a “protease interaction site”, but the longest isoform has a signal peptide (36 aa) and an “insulin growth factor-binding protein homologs” region. In chickens, there is a 311 aa IGFBP2 protein (containing four exons), and its structure is similar to the longest IGFBP2 isoform in humans. Geese, however, lack the longest IGFBP2 isoform, and have only two shorter isoforms (amino acids 185 and 189). Similar to the shorter isoforms of human IGFBP2, goose IGFBP2 isoforms only have the “thyroglobulin type-1 repeat” region and the “protease interaction site”. By aligning the 181 aa-long human IGFBP2 with 189 aa-long goose IGFBP2, we found that the identity between the two sequences was 76%, suggesting that goose and human IGFBP2 proteins may have similar functions.

IGFBP family members, including IGFBP2, constitute a system of insulin-like growth factor 1/2 (IGF1/2) and IGF receptors (IGF1R and IGF2R). IGFs have structures and functions similar to those of insulin, and their major functions include the promotion of growth and glucose uptake [9]. PI3K-AKT/PKB and Ras-MAPK pathways are activated [10]. IGFs can also bind to integrins to perform their functions. During the growth period in chicken, IGF1 is mainly synthesized in the liver [11]. IGFBP2 can bind to IGFs with a high affinity or remain intracellular and can interact with many different ligands. By binding to IGFs, IGFBP2 regulates their biological availability and prolongs their half-life. IGFBP2 plays an IGF-dependent or -independent role [12]. Previous studies have indicated that IGFBP2 is expressed in the liver, muscle, heart, ovaries, brain, intestine, and other chicken tissues [13]. Adipose tissue is the main site of IGFBP2 synthesis and secretion [14]. It is secreted via endocrine or paracrine mechanisms [15]. IGFBP2 can inhibit the biological effects of IGF in vivo, e.g., it can inhibit IGF-mediated growth and development. A previous study showed that nucleotide polymorphisms in the chicken *IGFBP2* intron are associated with growth traits [16]. IGFBP2 plays a key role in lipid metabolism. For example, the overexpression of *IGFBP2* in chicken liver cells can promote the expression of genes involved in fatty acid synthesis and increase the triglyceride content [17]. In addition to being indirectly regulated by the thyroid hormone, growth hormone, and leptin via IGF1 in chickens [18,19], the expression of *IGFBP2* can be regulated by nutritional status; fasting increases the expression of *IGFBP2*, whereas refeeding can reverse this increase in the chicken liver [20].

IGFBP2 plays an important role in animal nutrition and energy oversupply. For example, in a diet-induced obesity model, *IGFBP2* transgenic mice gained less body weight than the wild-type mice, suggesting that IGFBP2 prevents the development of obesity and insulin resistance [21]. The injection of IGFBP2 consistently improved the sensitivity of the mouse liver to insulin and significantly decreased the degree of hepatic steatosis. In the ob/ob mouse model, increased IGFBP2 levels significantly reduced the number of lipid droplets in the liver. Therefore, IGFBP2 might play an important regulatory role in hepatic lipid metabolism [22].

Compared with mammals, the biological effects caused by changes in nutritional or energy status in poultry, especially in geese, and the related mechanisms remain unclear. This study aims to explore the role and underlying mechanism of IGFBP2 in nutritional and energy metabolism of goose liver using the overfeeding and fasting/refeeding models. The findings from this study will not only shed light on the biological function of IGFBP2, but also provide insight into how to improve production efficiency by carefully regulating the nutritional status of geese.

## 2. Materials and Methods

### 2.1. Experimental Animals and Sample Collection

The Animal Care and Use Committee of Yangzhou University approved all of the animal experiments (permission number SYXK(Su)2021-0026). The same batch of 65-day-old healthy Landes male geese with similar body weights was fed at Licheng Livestock and Poultry Breeding Co., Ltd. (Huai’an, Jiangsu, China) for a 5-day preparatory overfeeding period before they were formally overfed for 24 days. The geese were assigned to two groups randomly, with one group (n = 8) having free access to feed and water as a control and the other being overfed. The geese were raised in cages in a semi-open house under natural light conditions. The overfeeding protocol has been described previously [23]. Briefly, the feed was administered via gavage. Feed intake during preparatory overfeeding was gradually increased from 100 to 300 g per day (twice on the first and second days and thrice on the third to fifth days). During the formal overfeeding, the total daily feed intake was 500 g three times a day for the first five days, followed by 800 g of daily feed (4 meals per day) for the following week, and 1200 g five times a day for the remaining days (from the 13th to 24th day of formal overfeeding). The diet consisted of cooked corn with 1% vegetable oil and 1% salt for both the overfed and control groups. At 93 days of age, eight geese were randomly selected from each group and fasted overnight with free access to water. After weighing and euthanasia with CO_2_ the next morning, the liver tissue samples were collected, snap-frozen in liquid nitrogen, and transferred to −80 °C for storage.

Healthy 7-day-old Landes goslings with similar body weights were obtained from the same batch and raised at Licheng Livestock and Poultry Breeding Co., Ltd. (Huai’an, Jiangsu, China). At 8 days old, the geese were randomly assigned to three groups (10 each): control, fasting, and refeeding. The control group had free access to food and water throughout the experiment. In the fasting group, the geese were without food for 24 h but had access to water. Similarly, the geese in the refeeding group also fasted for 24 h, but were then refed for 2 h with free access to food and water. The geese were all raised in a closed house with room temperature at 32 °C and 14 h of lighting time (06:00 a.m.–20:00 p.m.). The geese in the fasting group fasted from 09:00 a.m. to 09:00 a.m. the next day, and those in the refeeding group fasted from 07:00 a.m. to 07:00 a.m. the next day, followed by refeeding from 07:00 a.m. to 09:00 a.m. Eight goslings were randomly selected from each group and the selected ones were individually euthanized with CO_2_ in the following order: control, fasting, and refeeding groups. Once each bird was euthanized, the liver tissue samples were then collected, snap-frozen in liquid nitrogen, and transferred to −80 °C for storage.

### 2.2. Isolation and Culture of Goose Primary Hepatocytes

Primary hepatocytes were isolated from Landes geese embryos on the 23rd day of incubation. However, the sex of the embryos was not determined. The cells for each replicate were obtained from a single embryo. The cell isolation and culture methods have been described previously [23]. Briefly, several 23-day-old Landes-goose-embryonated eggs were removed from the incubator, and the egg surface was cleaned with 75% alcohol. The embryos were individually removed from the eggs and placed in Petri dishes. After the embryos were sacrificed, their livers were collected and rinsed thrice with physiological saline. Subsequently, the liver tissue was sheared and digested with 3–5 volumes of 0.2% type IV collagenase at 37 °C for 40 min (shake once per 10 min). An additional volume of complete medium (containing high glucose DMEM, 10% fetal bovine serum, 1% 100 IU/mL penicillin/streptomycin solution, and 10 μL 20 ng/mL EGF) was added to the dish to stop digestion. Next, cell clumps and undigested tissue were removed from the mixture using a 40 μm pore-sized cell strainer, which was followed by centrifugation at room temperature. After removing the supernatant, three volumes of red blood cell lysate were added to the cell pellet and the cells were suspended by gentle pipetting. The mixture was then placed in ice-cold water, incubated for 10 min, and centrifuged to remove the supernatant. The cell pellet was rinsed with three volumes of DMEM and centrifuged to remove the supernatant. The rinsing and centrifugation procedure was repeated. The cell pellet was suspended with a pre-warmed complete medium by gentle pipetting. After cell counting, 1 × 10^6^ cells per well were plated and cultured at 37 °C in a 5% CO_2_ incubator. The cells were cultured for 6 h and the medium was changed once. The cells were then prepared for the treatment assays.

### 2.3. Glucose and Insulin Treatments of Goose Primary Hepatocytes

The isolated goose primary hepatocytes were incubated in complete medium (high glucose DMEM, 10% fetal bovine serum, 1% 100 IU/mL penicillin−streptomycin solution, and 10 μL 20 ng/mL EGF) for 12 h, followed by treating the cells with different concentrations (0, 25, 50, and 100 mM) of glucose (Cat. No. G7021; Sigma-Aldrich, St. Louis, MO, USA) or different concentrations (0, 5, 10, and 20 nM) of insulin (Cat. No. I5500; Sigma-Aldrich, St. Louis, MO, USA) for 12 h. All of the culture media used in the treatment assays were complete media, except for those specifically indicated. Glucose and insulin supplemented in the medium were added to the assays. After 12 h of treatment, the cells were rinsed twice with physiological saline. Subsequently, 1 mL of TRIzol (Cat. No. DP424; Tiangen Biotech (Beijing, China) was added to each well. The cells were collected and stored at −80 °C for later use. There were six replicates for each treatment.

### 2.4. Overexpression and Interference Assays of IGFBP2 Gene in Goose Primary Hepatocytes

The IGFBP2 overexpression vector (GenePharma, Suzhou, China) was constructed according to the goose IGFBP2 reference sequence in GenBank (XM_013196707.2), and the coding sequence of the goose *IGFBP2* gene was inserted into the *Cytomegalovirus* (CMV) promoter-driven pcDNA3.1 vector. The *IGFBP2* overexpression vector and pcDNA3.1 empty vector were transfected into goose primary hepatocytes using Lipofectamine 2000 (Biosharp, Shanghai, China) according to the manufacturer’s instructions. After 6 h of transfection, Opti-MEM (Thermo Fisher Scientific, Waltham, MA, USA) was replaced with a complete medium. To prepare the samples for quantitative PCR (qPCR) analysis, the cells were incubated for 24 h. The cells were rinsed twice with physiological saline before collection by centrifugation. Each experimental group comprised six replicates.

For the interference assay of *IGFBP2* in goose primary hepatocytes, siRNA was first designed and synthesized by GenePharma (Suzhou, China) according to the reference sequence of *IGFBP2* mRNA. The sense strand sequence of siRNA targeting *IGFBP2* (*IGFBP2*-siRNA) is 5′-UGGAACGCAUCUCCACCAUTT-3′. The antisense strand sequence is 5′-AUGGUGGAGAUGCGUUCCATT-3′. *IGFBP2*-siRNA and scrambled negative control siRNA (NC-siRNA) were transfected into goose primary hepatocytes using Lipofectamine 2000 (Biosharp, Shanghai, China) according to the manufacturer’s instructions. After 6 h of transfection, Opti-MEM was replaced with complete medium, and the cells were collected after 24 h of incubation. The interference effect of *IGFBP2*-siRNA was verified by quantitative PCR (qPCR) analysis. Subsequently, *IGFBP2*-siRNA and NC-siRNA were used in the following assays: Briefly, the isolated cells were divided into three groups, i.e., control group (10% serum + glucose-free DMEM + NC-siRNA), treatment group 1 (2% serum + glucose-free DMEM + NC-siRNA), and treatment group 2 (2% serum + glucose-free DMEM + *IGFBP2*-siRNA). Glucose-free DMEM was purchased from Sigma-Aldrich (Cat. No. D5030; St. Louis, MO, USA). For the interference assay, goose primary hepatocytes were first transfected with NC-siRNA and *IGFBP2*-siRNA for 6 h using Lipofectamine 2000 (Biosharp, Shanghai, China) according to the manufacturer’s instructions, and then the medium was replaced with the corresponding medium (including 10% serum + glucose-free DMEM for the control group and 2% serum + glucose-free DMEM for the treatment group). The cells were then rinsed twice with physiological saline and 1 mL TRIzol was added. After complete lysis of the cells, the cell lysate was collected and stored at −80 °C for later use. Each group comprised six replicates.

### 2.5. Total RNA Isolation and qRT-PCR Analysis

RNA isolation and purification, cDNA synthesis by reverse transcription, and qPCR were performed as previously described [24]. Briefly, the total RNA was isolated from the liver tissue or cell samples using TRIzol reagent according to the manufacturer’s instructions. The quality and quantity of the total RNA were assessed using a NanoDrop 1000 spectrophotometer (Thermo Scientific, Wilmington, DE, USA). For cDNA synthesis, a total of 500 ng of RNA per sample and the mixture of random primers and oligo dT primer mix in a 20 μL reaction system were used to reverse-transcribe cDNA with the HiscriptTM Q RT Supermix Reverse Transcription Kit (Cat. No. R123-01; Vazyme Biotech Co., Ltd., Nanjing, China). The synthesized cDNA samples were then diluted five times before qPCR analysis. According to the manufacturer’s instructions for the Vazyme ACEQ QPCR SYBR Green Master Mix Kit (Novozyme Biotech Co. Q111-02/03; Vazyme Biotech Co., Ltd.), 4 pmol per primer and 1 μL cDNA sample were added to a 20 μL volume of the reaction system, and qPCR was performed on an ABI 7500 real-time quantitative PCR system (Applied Biosystems, Foster City, CA, USA). The reaction conditions were as follows: 95 °C for 5 min, followed by 40 cycles of 95 °C for 10 s and 60 °C for 30 s. Three technical replicates were set up for each sample, with an internal reference gene, which was β-actin. The primers used for qPCR were designed according to their corresponding reference sequences in GenBank using Primer 3.0 software. The primer sequences are listed in Appendix A. The mRNA expression level of the gene of interest relative to that of the internal control gene was calculated using the 2^−ΔΔCt^ method [25].

### 2.6. Transcriptome Analysis

The procedures for the transcriptome analysis of cells transfected with the *IGFBP2* overexpression vector or empty vector (control) have been described previously [26]. Briefly, mRNA was obtained from the total RNA through poly T oligo-attached magnetic beads enrichment (Sigma-Aldrich, St. Louis, MO, USA). RNA quality was then assessed for degradation and contamination, RNA purity measured using a NanoPhotometer^®^ spectrophotometer (IMPLEN, Westlake Village, CA, USA), and RNA integrity was evaluated using the RNA Nano 6000 Assay Kit of the BioAnalyzer 2100 system (Agilent Technologies, Santa Clara, CA, USA). Then, 1 μg RNA per sample was used to construct the cDNA library with NEBNEXT^®^ Ultra^TM^ RNA Library Prep Kit for Illumina (Illumina, San Diego, CA, USA) according to the manufacturer’s instructions. Different index codes were used to label the sequences for each sample. After library construction, initial quantification was performed using a Qubit 2.0 Fluorometer (Thermo Fisher Scientific, Waltham, MA, USA). The insert size in the library was detected using an Agilent 2100 Bioanalyzer (Agilent, Santa Clara, CA, USA). When the insert size reached the expected value, the effective library concentration was accurately quantified using real-time qPCR (the effective library concentration should be higher than 2 ng/μL) to ensure the library quality.

The constructed libraries were then sequenced using the Illumina NovaSeq platform (San Diego, CA, USA). Raw sequencing data were cleaned by removing adaptor sequences, poly N-containing sequences, and low-quality reads. Clean reads were analyzed using Phred software to calculate the Q20, Q30, and GC contents. Clean readings above Q30 were selected for further analysis. The Anser cygnoides domestic genome assembly in GenBank was used as the reference genome sequence (Anser cygnoides Goose V1.0). Clean data were annotated using HISAT2. Unigenes were acquired using the StringTie software. The gene expression levels were quantified using the FPKM (fragments per kilobase of transcript per million mapped reads) normalization method. Differentially expressed genes (DEGs) were identified by calculating and comparing the expression of each gene across groups using DESeq software [27]. The *p*-values were adjusted using the Benjamin−Hochberg method. The criteria for DEGs were a fold-change of the treatment group over the control group > 2 or <0.5, and a corrected *p*-value (adjusted *p*) < 0.05. Using the identified DEGs, gene ontology (GO) enrichment analysis and KEGG pathway enrichment analysis were performed using the ClusterProfiler R package [28]. The sequencing data were submitted to the GenBank database (https://www.ncbi.nlm.nih.gov/bioproject/PRJNA945445) under accession number PRJNA945445 (the access date is 16 March 2023).

### 2.7. Statistical Analysis

Data are presented as the mean ± SEM. Differences between two groups were analyzed using an unpaired two-tailed *t*-test, while the differences between more than two groups were analyzed using one-way analysis of variance (ANOVA), followed by multiple comparisons using Duncan’s method. Prior to analysis, statistical significance was set at *p* < 0.05. analysis.

## 3. Results

### 3.1. Effect of Changes in Nutritional Status on IGFBP2 Expression in Goose Liver

To clarify whether IGFBP2 mediates the biological effects caused by changes in nutritional status in the goose liver, overfeeding and fasting/refeeding models were used in this study. Quantitative PCR analysis showed that in the overfeeding model, overfeeding resulted in significant inhibition of mRNA expression levels of *IGFBP2* in the liver compared with the normally-fed control group (Figure 1A). In the fasting/refeeding model, fasting significantly induced the mRNA expression of *IGFBP2* in the liver compared with the normally-fed control, and this induction was reversed by refeeding (Figure 1B). These findings suggest that *IGFBP2* expression in the goose liver is influenced by changes in nutritional status, suggesting that IGFBP2 may play a role in mediating the biological effects associated with changes in energy levels in the liver.

### 3.2. Effect of Glucose and Insulin on IGFBP2 Expression in Goose Primary Hepatocytes

To determine the energy-related factors affecting *IGFBP2* expression, various concentrations of glucose and insulin were used to treat the primary hepatocytes from geese. Quantitative PCR analysis indicated that glucose had no significant effect on *IGFBP2* mRNA levels in the geese primary hepatocytes (Figure 2A), whereas insulin significantly inhibited *IGFBP2* mRNA levels in the geese primary hepatocytes (Figure 2B). These results suggest that the effect of changes in nutritional status on *IGFBP2* expression in vivo may be partially attributable to changes in insulin concentration.

### 3.3. Downstream Genes and Pathways Affected by IGFBP2 Overexpression

Transfection with the *IGFBP2* overexpression vector led to a significant increase in mRNA level of *IGFBP2* compared with the cells transfected with the control vector, pcDNA3.1(+), as determined by quantitative PCR analysis (Figure 3A), i.e., the *IGFBP2* gene was successfully overexpressed in the goose primary hepatocytes. After sequencing, a total of 56.8 G base pairs of sequences were acquired with an average of 7.1 G, and the mapping rates ranged from 66.7~73.6% (Appendix A). Transcriptome analysis of the cells identified 337 DEGs, including 111 with an up-regulated expression and 226 with a down-regulated in the cells transfected with the *IGFBP2* overexpression vector vs. the control cells, are listed in Attachment S1. The top 10 most significantly (based on *p*-value) up- and down-regulated DEGs are listed in Table 1. The top 10 up- and down-regulated DEGs with the largest |log2(fold change)| are listed in Appendix A. KEGG pathways significantly enriched with DEGs were also identified, which mainly included cytokine−cytokine receptor interaction, cytosolic DNA-sensing pathway, toll-like receptor signaling pathway, AGE-RAGE signaling pathway in diabetic complications, arachidonic acid metabolism, metabolism of xenobiotics by Cytochrome P450, focal adhesion, NOD-like receptor signaling pathway, and other pathways (Figure 3B). The DEGs enriched in these KEGG pathways are listed in Appendix A. Some randomly selected DEGs were validated by qPCR analysis, and the results showed that except for *RSAD1* and *FGB* whose expression was not significantly affected by *IGFBP2* overexpression, the DEGs including *MAP3K7CL*, *SLC4041R1*, *LOC106038123*, *LOC106041089*, *STAT4*, *IL22RA2*, *IL6*, *EX-FABP COTL1*, and *CD9* were all significantly affected by *IGFBP2* overexpression (Figure 3C), suggesting that the results of the transcriptome analysis were reliable.

### 3.4. Effect of IGFBP2 Overexpression on the Expression of the Genes in the Cytokine−Cytokine Receptor Pathway

The qPCR analysis of 10 DEGs in the cytokine−cytokine receptor pathway showed that, except for *LOC106038554*, whose mRNA expression was not significantly affected by *IGFBP2* overexpression, the genes including *LOC106041919*, *CCL20*, *IL6*, *LOC106042256*, *LOC106041041*, *IL22RA1*, *LOC106038123*, *LOC106041089*, and *LOC106041040* were significantly inhibited by *IGFBP2* overexpression (Figure 4). This suggests that IGFBP2 may exert its biological role by inhibiting the cytokine−cytokine receptor pathway.

### 3.5. Effect of IGFBP2 Interference on the Expression of the Genes in the Cytokine−Cytokine Receptor Pathway

Based on the mRNA sequence of the goose *IGFBP2* gene, an siRNA was designed against *IGFBP2* and goose primary hepatocytes transfected with *IGFBP2*-siRNA or negative control siRNA (NC-siRNA) were used for the qPCR analysis. According to the results, the mRNA expression of *IGFBP2* was significantly reduced in the goose primary hepatocytes transfected with *IGFBP2*-siRNA by 62% compared with those transfected with the control NC-siRNA. This suggests that *IGFBP2*-siRNA effectively down-regulated the expression of *IGFBP2* in the cells (Figure 5A). Furthermore, qPCR was performed to determine the expression of the DEGs in the cytokine−cytokine receptor pathway described above (*LOC106041919*, *CCL20*, *IL6*, *LOC106042256*, *LOC106041041*, *IL22RA1*, *LOC106038123*, *LOC106041089*, and *LOC106041040*), and the data indicated that the mRNA expression in *LOC106041919*, *CCL20*, *IL6*, *LOC106042256*, *LOC106041041*, *IL22RA1*, and *LOC106041089* genes was significantly increased after *IGFBP2* interference (Figure 5B), which is in contrast with the suppression of the genes by *IGFBP2* overexpression, thus further validating that IGFBP2 regulates these genes and the related cytokine−cytokine receptor pathway.

### 3.6. Effect of Overfeeding on the Expression of the Genes in the Cytokine−Cytokine Receptor Pathway in Goose Liver

The quantitative PCR analysis showed that overfeeding resulted in a significant increase in the mRNA expression of *LOC106041919*, *CCL20*, *IL6*, *LOC106042256*, *IL22RA1*, and *LOC106038123* in the goose liver compared with the normally fed control (Figure 6), suggesting that IGFBP2 may be involved in regulating the expression of these genes during overfeeding.

### 3.7. Effect of Fasting and Refeeding on the Expression of the Genes in the Cytokine−Cytokine Receptor Pathway in Goose Liver

The data from the qPCR analysis showed that fasting resulted in a significant decrease in the mRNA expression of *LOC106041919*, *CCL20*, *LOC106042256*, *LOC106041041*, *IL22RA1*, and *LOC106038123* genes in the goose liver compared to the normally fed control, whereas refeeding reversed the changes in the expression of these genes with the induction of *LOC106041919*, *LOC106042256*, *IL22RA1*, and *LOC106038123* genes, reaching statistical significance (Figure 7). These findings suggest that IGFBP2 may be involved in the regulation of the expression of these genes during fasting and refeeding.

### 3.8. Validation of IGFBP2 Regulating the Expression of the Genes in the Cytokine−Cytokine Receptor Pathway

To further validate the regulation of IGFBP2 on the expression of genes in the cytokine−cytokine receptor pathway, we constructed a cell model with elevated *IGFBP2* expression using low serum (2%, low insulin content) and glucose-free medium. Subsequently, this model was used to verify the findings that the overexpression of *IGFBP2* inhibited the expression of genes in the cytokine−cytokine receptor pathway. The *IGFBP2* interference assay indicated that the induction of *IGFBP2* expression by low serum medium was significantly suppressed by *IGFBP2*-siRNA (Figure 8A). Correspondingly, the decrease in the expression of genes (including *CCL20*, *LOC106042256*, and *LOC106038123*) in the cytokine−cytokine receptor pathway by the low serum medium was also inhibited by *IGFBP2* siRNA (Figure 8B). These findings further confirmed that the expression of *CCL20*, *LOC106042256*, and *LOC106038123* genes in the cytokine−cytokine receptor pathway was regulated by IGFBP2.

## 4. Discussion

The liver plays a vital role in the regulation of animal nutritional metabolism, and its metabolism and physiology are easily influenced by changes in nutritional status. In recent years, a number of studies have revealed this influence through transcriptome analysis. For example, transcriptomic analysis of the livers of 4-week-old broilers starved for 48 h or the *ad libitum* feeding broilers reveals that the genes related to fatty acid oxidation, ketone synthesis, and gluconeogenesis (e.g., *ALDOB*, *LDHB*, and *LPIN2*) are induced, and the genes related to fatty acid and cholesterol synthesis (e.g., *FASN*, *ME1*, and *SCD*) are suppressed by fasting [29]. Consistently, transcriptome analysis of the liver in newly-hatched chicks indicates that the lipogenic genes are acutely depressed by fasting, but elevated by refeeding, while the lipolytic genes are up-regulated by fasting or suppressed by refeeding [30]. Moreover, it has been found that intermittent fasting is associated with large and reciprocal changes in the genes related to lipid and carbohydrate metabolism, but also chronic changes in the genes related to amino acid metabolism (generally down-regulated) and cell cycle progression (generally up-regulated), as well as a small number of inflammation-related genes [31]. Interestingly, transcriptomic analyses of goose livers have indicated that fasting mainly affects signaling pathways related to lipid metabolism, whereas refeeding affects not only lipid metabolism pathways, but also glucose and amino acid metabolism pathways [32]. The peroxisome proliferator-activated receptor (PPAR) signaling pathway may play a significant role in lipid metabolism. Together, these findings suggest that changes in nutritional status have significant impacts on liver metabolism and physiology at a molecular level. However, further studies are needed to address how these genes are regulated mechanistically.

IGFBP2 plays important roles in cell growth, proliferation, and differentiation as a growth factor (IGF)-binding protein. Previous studies have indicated that IGFBP2 regulates cell growth, proliferation, and differentiation through the TGF-β/SMAD signaling pathway and controls the movement and apoptosis of cells [11]. IGFBP2 also participates in cell growth, proliferation, and metabolism through the PI3K/AKT signaling pathway [33]. Upon the binding of IGF1 to its receptor (IGF1R), PI3K is activated to promote cell proliferation by activating MAPK and preventing apoptosis by inactivating pro-apoptotic proteins (e.g., BAD) [34]. In the present study, the transcriptome analysis of goose hepatocytes transfected with *IGFBP2* overexpression vector versus empty vector indicated that IGFBP2 could play a biological role through the cytokine−cytokine receptor pathway and that the overexpression of *IGFBP2* inhibited the expression of many genes in this pathway. Cytokines, as secreted proteins, are involved in cell growth, differentiation, and death, as well as in immune/inflammatory responses upon binding to their respective receptors. For example, the cytokines IL22RA1 [35] and IL6 [36] promote cell growth through the JAK/STAT signaling pathway. IL-17 [37] and CCL20 [38] also promote cell growth and proliferation. Interestingly, the expression level of IGFBP2 in goose liver was negatively correlated with the expression levels of these cytokines in the fasting/refeeding or overfeeding models, which was in line with the results of the *IGFBP2* overexpression and interference assays. These results suggest that *IGFBP2* participates in modulating the growth and development of goose liver through the cytokine−cytokine receptor pathway in response to nutritional or energy deficiency or surplus. It is known that IGFs can regulate cell growth, proliferation, and cell death through activation of the TGF-β/SMAD signaling pathway [39], PI3K-AKT/PKB signaling pathway [40], and the Ras-MAPK signaling pathway [41]; IGFBP2 can affect the bioavailability of IGFs by binding to IGFs [42]; both fasting and refeeding can influence the growth of muscle and liver and the expression of *IGFBP2* in the liver, while overfeeding for 19 days can influence the expression of genes in the cell growth and death pathway in goose liver [43]. Therefore, the role of IGFBP2 in growth and development may be mediated by the IGF/IGFR system in the fasting/refeeding and overfeeding models.

Cytokines play important roles in inflammation and immune responses. In the present study, *IGFBP2* overexpression affected the expression of genes involved in the Toll-like receptor, NOD-like receptor, and AGE-RAGE signaling pathways in diabetic complications, in addition to the cytokine−cytokine receptor pathway. These pathways are associated with immunity and inflammation, suggesting that IGFBP2 may also participate in the modulation of inflammation and immune responses. In line with this notion, previous studies have indicated that chronic changes induced by intermittent fasting can affect the expression of some inflammation-related genes in chicken liver [44], and overfeeding for 19 days can influence the expression of genes in the immune disease pathway in goose fatty liver [26]. These findings together suggest that hepatic IGFBP2 mediates the effects of changes in nutritional status on the expression of inflammatory cytokines via the cytokine–cytokine receptor pathway.

Furthermore, this study also found that *IGFBP2* overexpression affected the expression of genes in the pathways related to lipid metabolism. These pathways include arachidonic acid metabolism, primary bile acid biosynthesis, fatty acid metabolism, cholesterol metabolism, unsaturated fatty acid biosynthesis, and linoleic acid metabolism. Previous studies have consistently indicated that adipose tissue is the main site of IGFBP2 synthesis and secretion, and that the overexpression of *IGFBP2* in chicken liver cells can promote the expression of genes involved in fatty acid synthesis and increase the triglyceride content [17]. These findings suggested that IGFBP2 plays a role in lipid metabolism. As mentioned above, the transcriptome analysis of poultry liver in the fasting/refeeding model and the overfeeding model indicates that changes in nutritional status affect the expression of genes related to lipid metabolism in the liver [31], i.e., fasting induces the up-regulation of genes related to lipid oxidation and down-regulation of genes related to fatty acid synthesis in chicken liver, and vice versa for refeeding [45]; we speculate that IGFBP2 may mediate the effects of nutritional status changes on the expression of lipid metabolism-related genes.

## 5. Conclusions

In conclusion, IGFBP2 is involved in the regulation of lipid metabolism, cell growth, differentiation, and immunity/inflammation via changes in nutritional status mainly through the cytokine−cytokine receptor pathway and other pathways (Figure 9).

## Figures and Tables

**Figure 1 animals-13-02336-f001:**
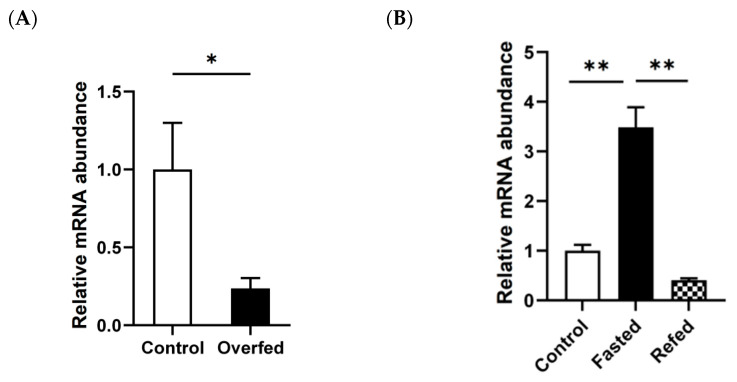
The mRNA expression levels of *IGFBP2* in goose liver were correlated with nutritional status. (**A**) Overfeeding significantly inhibited the mRNA expression of *IGFBP2* in goose liver. (**B**) The mRNA expression of *IGFBP2* in the liver of geese was induced by fasting, and this induction was inhibited by refeeding. *, ** denote *p* < 0.05 and *p* < 0.01 for the fasting group vs. the control group, or the refeeding group vs. the fasting group, respectively. The fasted group was fasted for 24 h, the refed group was fasted for 24 h, followed by 2 h of refeeding, and the control group was normally fed with free access to feed and water. The mRNA expression of *IGFBP2* was determined by qPCR while the protein expression was determined by immunoblot analysis. n = 8. The internal reference gene was β-actin and all data are presented as the mean ± SEM.

**Figure 2 animals-13-02336-f002:**
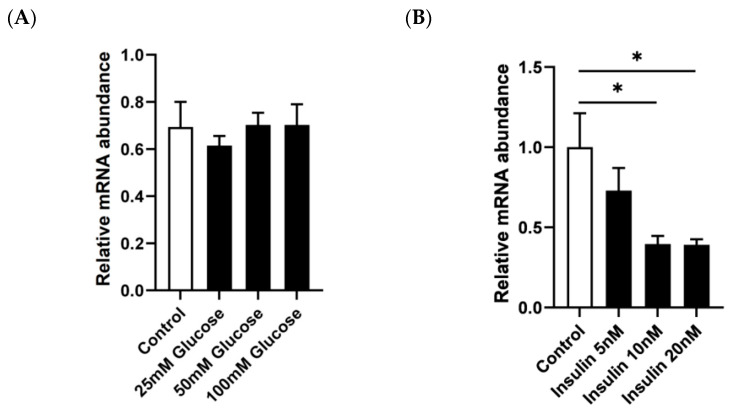
The mRNA expression level of *IGFBP2* in goose primary hepatocytes were treated with different concentrations of glucose (**A**) or insulin (**B**) versus the control. Note: The mRNA expression of *IGFBP2* was determined by qPCR. The control hepatocytes were not treated with nutritional factors. * denotes *p* < 0.05 versus control, respectively. n = 6. The internal reference gene was β-actin and all data are presented as the mean ± SEM.

**Figure 3 animals-13-02336-f003:**
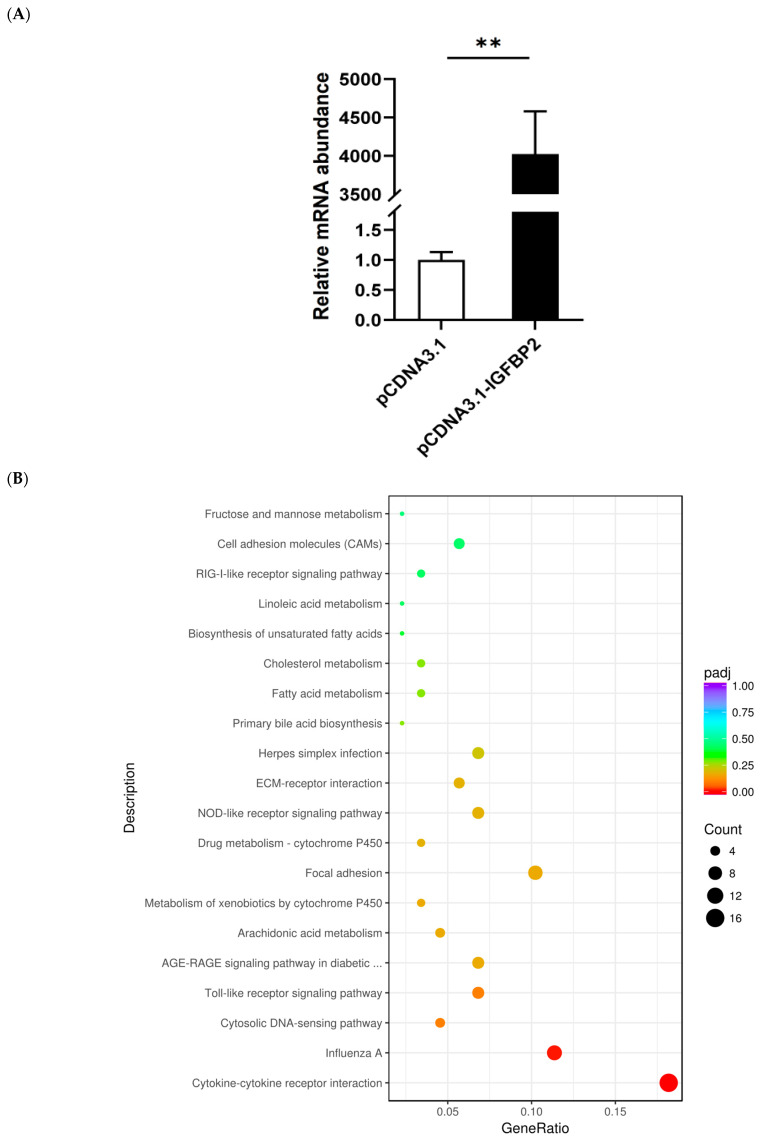
Transcriptome analysis on goose primary cells transfected with *IGFBP2* overexpression vectors and empty vectors. (**A**) Quantitative PCR showed that *IGFBP2* mRNA expression was significantly higher in the cells transfected with *IGFBP2* overexpression vectors than in the control group. (**B**) KEGG enrichment analysis of DEGs identified in this study was displayed using a dot chart, with the horizontal axis indicating the ratio of annotated DEGs on a specific KEGG pathway to the total number of annotated DEGs, and the vertical axis indicating the KEGG pathways. The adjusted *p*-value (*p*-adj) was represented by color-coding, while the number of DEGs was denoted by the dot size. The results indicate a significant difference between goose primary hepatocytes transfected with *IGFBP2* overexpression vectors and empty vectors. The KEGG pathways enriched with the up-regulated or down-regulated DEGs are shown in Appendix A. (**C**) DEGs from the transcriptome analysis were randomly selected and validated using quantitative PCR in goose primary hepatocytes transfected with either the *IGFBP2* overexpression vector or the empty vector control group. The mRNA expression of the genes of interest was determined by qPCR, while the protein expression was determined by immunoblot analysis. *, ** denote *p* < 0.05 and *p* < 0.01 versus the control, respectively. n = 6. The internal reference gene was β-actin and all data are presented as the mean ± SEM.

**Figure 4 animals-13-02336-f004:**
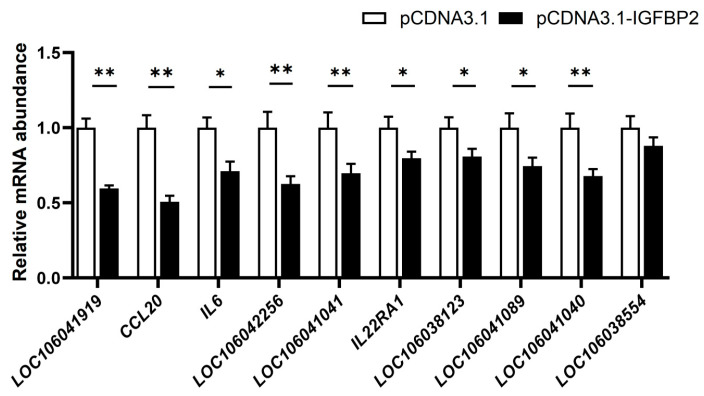
The mRNA expression level of the DEGs in the cytokine−cytokine receptor pathway in goose primary hepatocytes transfected with *IGFBP2* overexpression vector versus empty vector (the control). The mRNA expression of the genes of interest was determined by qPCR. *, ** denote *p* < 0.05 and *p* < 0.01 versus the control, respectively. n = 6. The internal reference gene was β-actin and all data are presented as the mean ± SEM.

**Figure 5 animals-13-02336-f005:**
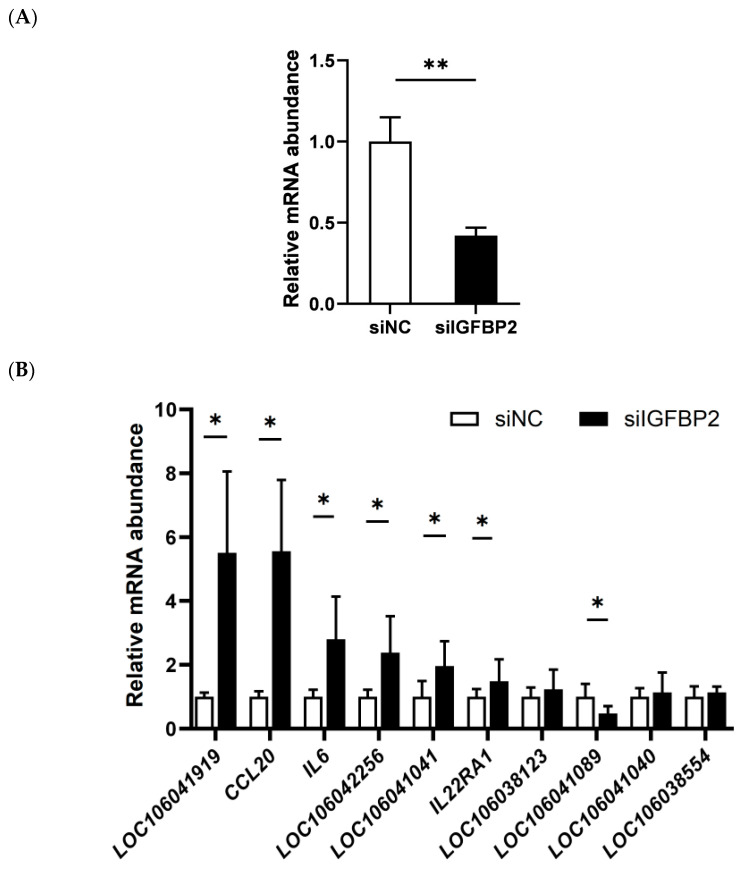
The mRNA expression level of *IGFBP2* (**A**) and the representative DEGs (**B**) in the cytokine−cytokine receptor pathway in goose primary hepatocytes transfected with siRNA targeting to *IGFBP2* (siIGFBP2) versus the negative control siRNA (siNC). The mRNA expression of the genes of interest was determined by qPCR. *, ** denote *p* < 0.05 and *p* < 0.01 versus the control (siNC), respectively; n = 6. The internal reference gene was β-actin and all data are presented as the mean ± SEM.

**Figure 6 animals-13-02336-f006:**
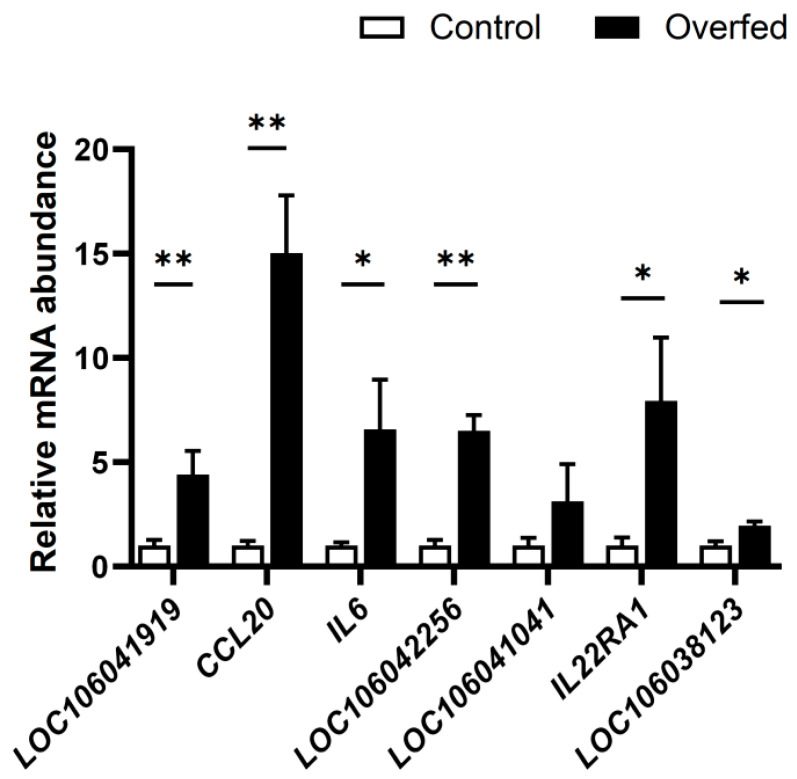
The mRNA expression levels of representative DEGs in the cytokine−cytokine receptor pathway were measured in the livers of overfed and control geese. The overfed geese were overfed for 24 days while the control geese had free access to feed and water. The gene expression was measured using qPCR. *, ** denote *p* < 0.05 and *p* < 0.01 versus the control, respectively; n = 8. The internal reference gene was β-actin and all data are presented as the mean ± SEM.

**Figure 7 animals-13-02336-f007:**
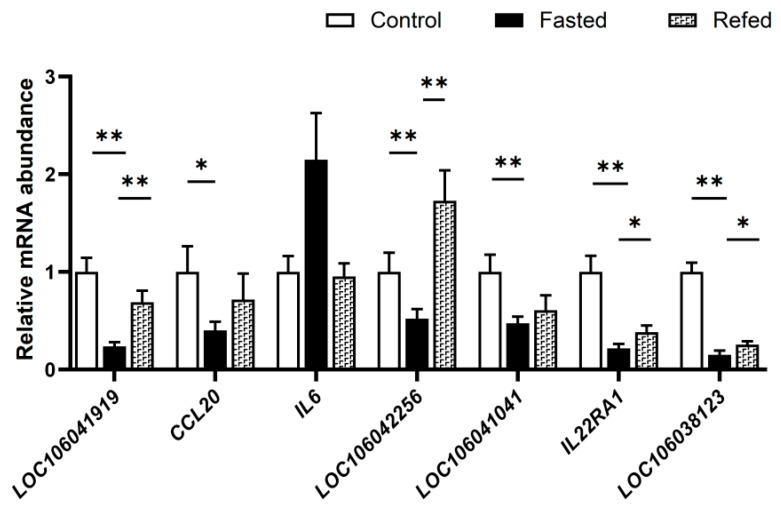
The mRNA expression levels of DEGs in the cytokine−cytokine receptor pathway were analyzed in the livers of the control, fasted, and refed geese. The fasted group was fasted for 24 h, the refed group was fasted for 24 h and then refed for 2 h, and the control group had free access to feed and water. *IGFBP2* mRNA levels were measured using qPCR. *, ** denote *p* < 0.05 and *p* < 0.01 for the fasting group vs. the control group, or the refeeding group vs. the fasting group, respectively. n = 8. The internal reference gene was β-actin and all data are presented as the mean ± SEM.

**Figure 8 animals-13-02336-f008:**
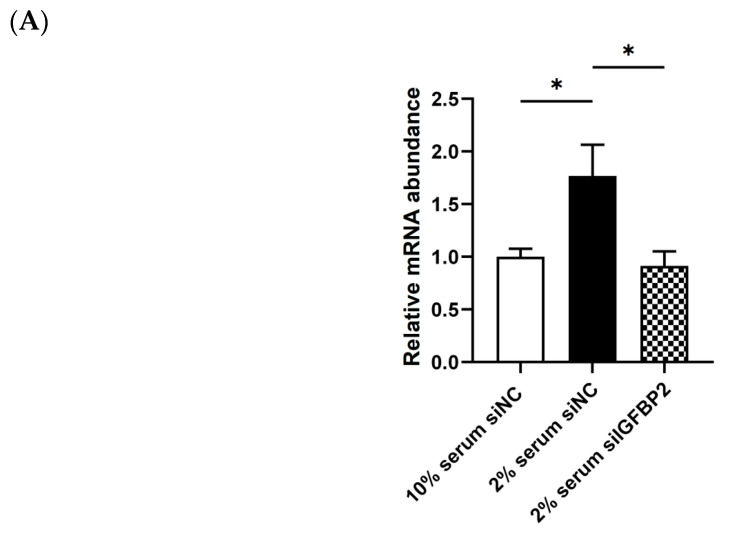
The mRNA expression levels of *IGFBP2* (**A**) and the representative DEGs (**B**) in the cytokine−cytokine receptor pathway in goose primary hepatocytes transfected with *IGFBP2* siRNA (siIGFBP2) vs. the negative control siRNA (siNC). The mRNA expression of the genes of interest was determined by qPCR. Note: 10% serum medium includes glucose-free DMEM supplemented with 10% serum; 2% serum medium includes glucose-free DMEM supplemented with 2% serum. * denotes *p* < 0.05 for the Medium1 siNC vs. Medium2 siNC, or the Medium2 siNC vs. Medium2 siIGFBP2. n = 6. The internal reference gene was β-actin and all data are presented as the mean ± SEM.

**Figure 9 animals-13-02336-f009:**
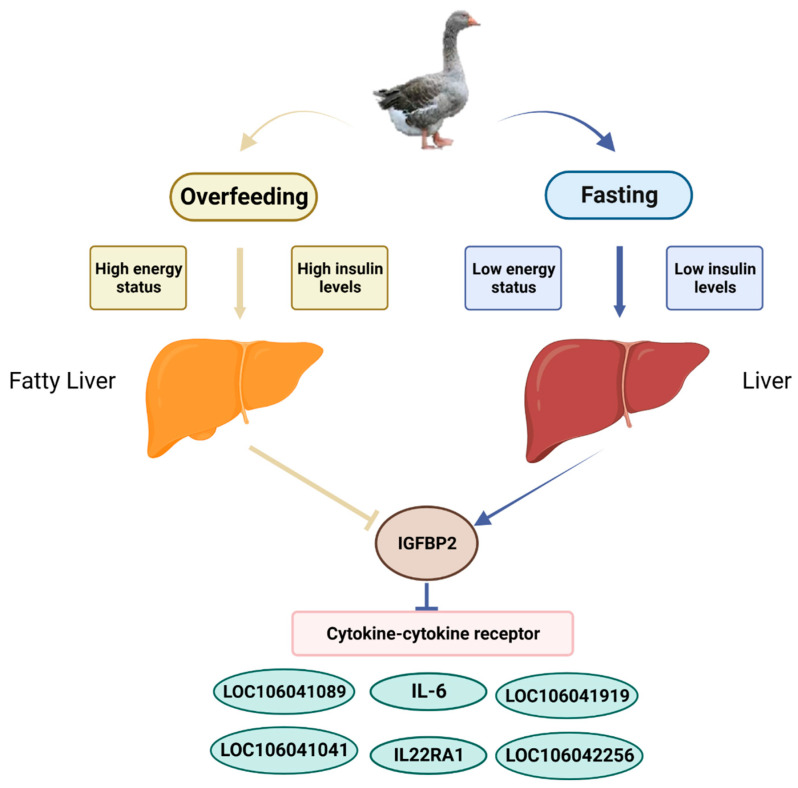
A sketch illustrating the regulation of the IGFBP2 expression by changes in nutritional status and the mechanism of IGFBP2 participating in growth and inflammation caused by the changes. When geese are overfed with an energy-rich diet, the concentration of insulin in the blood is elevated, which inhibits hepatic the IGFBP2 expression, and this inhibition can influence the expression of genes in the cytokine−cytokine receptor signaling pathway, and vice versa when geese are fasted.

**Table 1 animals-13-02336-t001:** Top 10 up- and down-regulated differentially expressed genes (sorted by adjusted *p*-value).

Gene Name	Gene Symbol	Log_2_ (Fold Change)	*p*-Value	Adjusted *p*-Value
Up-regulated				
Radical S-Adenosyl Methionine Domain Containing 2	*RSAD2*	0.93	1.07 × 10^−31^	3.01 × 10^−28^
Solute Carrier Family 40 Member 1	*SLC40A1*	0.85	4.03 × 10^−19^	3.24 × 10^−16^
Sorbin And SH3 Domain Containing 2	*SORBS2*	1.14	4.56 × 10^−13^	1.77 × 10^−10^
Fibrinogen Beta Chain	*FGB*	0.57	1.95 × 10^−12^	6.67 × 10^−10^
Lecithin-Cholesterol Acyltransferase	*LCAT*	0.69	2.64 × 10^−12^	8.49 × 10^−10^
Monoacylglycerol O-Acyltransferase 1	*MOGAT1*	0.56	7.49 × 10^−11^	1.72 × 10^−8^
Fibrinogen Gamma Chain	*FGG*	0.47	4.67 × 10^−10^	9.56 × 10^−8^
Collectin Subfamily Member 10	*COLEC10*	1.34	7.46 × 10^−10^	1.50 × 10^−7^
Adenomatosis Polyposis Coli Down-Regulated 1 Protein	*APCDD1*	0.53	4.33 × 10^−9^	7.74 × 10^−7^
D-Dopachrome Tautomerase	*LOC106031299*	0.48	7.94 × 10^−9^	1.35 × 10^−6^
Down-regulated				
Uncharacterized LOC106034664	*LOC106034664*	−1.65	1.56 × 10^−43^	8.79 × 10^−40^
C-C Motif Chemokine 5-like	*LOC106041040*	−1.36	2.44 × 10^−38^	9.17 × 10^−35^
Cystatin-Like	*LOC106044772*	−1.05	3.99 × 10^−29^	7.50 × 10^−26^
Toll-like Receptor 2	*LOC106042256*	−2.07	4.86 × 10^−28^	7.81 × 10^−25^
Extracellular Fatty Acid-binding Protein-like	*LOC106037025*	−1.01	1.15 × 10^−25^	1.62 × 10^−22^
Coactosin Like F-Actin Binding Protein 1	*COTL1*	−0.70	1.11 × 10^−24^	1.39 × 10^−21^
Solute Carrier Family 13 Member 5	*SLC13A5*	−1.50	1.29 × 10^−23^	1.46 × 10^−20^
Protein MRP-126	*LOC106049124*	−0.75	5.03 × 10^−23^	5.16 × 10^−20^
Chimerin 2	*CHN2*	−0.87	1.60 × 10^−22^	1.50 × 10^−19^
CD9 Molecule	*CD9*	−1.05	1.86 × 10^−20^	1.61 × 10^−17^

Note: To identify differentially expressed genes (DEGs) in goose primary hepatocytes transfected with *IGFBP2* overexpression vectors versus the empty vector, transcriptome analysis was performed. By comparing the gene expression profiles of the two groups, the DEGs were identified. The genes shown in this table are the top 10 up-regulated or down-regulated DEGs with the lowest adjusted *p*-value and |log2(fold change)| > 1. The fold change refers to the ratio of the expression level of the DEG in the *IGFBP2*-overexpressing cells to that of the control cells. The letter ‘E’ indicates the scientific notation.

## Data Availability

The sequencing data from this study were submitted to the GenBank databases (https://www.ncbi.nlm.nih.gov/bioproject/PRJNA945445) and assigned accession number PRJNA945445 (the access date is 16 March 2023).

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
