# Peer review of "Goose Hepatic IGFBP2 Is Regulated by Nutritional Status and Participates in Energy Metabolism Mainly through the Cytokine−Cytokine Receptor Pathway"

_animals, 2023, doi:10.3390/ani13142336_

Round 1
Reviewer 1 Report
Li et al.'s manuscript proposes that IGFBP2 mediates the biological effects caused by alterations in nutritional or energy levels via the cytokine-cytokine receptor pathway. The study is well-executed, and the authors utilized various methods such as bioinformatics, in vitro, and in vivo experiments. Although the manuscript is intriguing, I have a few significant comments:
1. The introduction section contains ample space for elaborating on the transcriptomic results, and it should be trimmed appropriately and placed in the discussion section.
2. It is unclear why IGFBP2-overexpressing goose primary hepatocytes were selected for transcriptome sequencing analysis instead of directly sequencing the liver of control and fasted groups. Since fasting significantly induced the mRNA expression of IGFBP2 in the liver compared to the normally-fed control, it would be beneficial to explain the reasoning behind this choice.
3. In Figure 9, the authors concluded that IGFBP2 affects growth and inflammation, but there is no production data mentioned in the full text. Therefore, the authors should consider whether this is an appropriate graph.
4. It is unclear how to explain the difference in the effects of overexpression of IGFBP2 (Fig.4) and siIGFBP2 (Fig.5B) on the Cytokine-Cytokine Receptor Pathway. Although the results suggest that siIGFBP2 has a stronger effect on certain gene promotion (LOC106041919 and CCL20), overexpression has a more substantial influence than siIGFBP2 on gene expression (Fig.3A) than siIGFBP2 (Fig.5A).
5. The clarity of Figure 3B is insufficient. Please improve the quality of the image.
6. References 2, 20, 25, 38, 39, 45, 51, and 52 have formatting issues such as incomplete page numbers or missing volume numbers, and they should be corrected according to American English standards.
English needs improvement.
Author Response
Reviewer #1:
Comments and Suggestions for Authors
Li et al.'s manuscript proposes that IGFBP2 mediates the biological effects caused by alterations in nutritional or energy levels via the cytokine-cytokine receptor pathway. The study is well-executed, and the authors utilized various methods such as bioinformatics, in vitro, and in vivo experiments. Although the manuscript is intriguing, I have a few significant comments.
Response: Thank you very much for your positive comments and constructive suggestions, which help us to improve this manuscript significantly. We have carefully read and consider your comments and suggestions, and made revisions to the manuscript accordingly. The point by point responses are shown as follows:
Comment 1: The introduction section contains ample space for elaborating on the transcriptomic results, and it should be trimmed appropriately and placed in the discussion section.
Response: Thank you very much for your constructive suggestions. We have revised the Introduction section to make it conciser and moved the transcriptomic results to the Discussion section accordingly. Please see in Line 70-72 and 471-492.
Comment 2: It is unclear why IGFBP2-overexpressing goose primary hepatocytes were selected for transcriptome sequencing analysis instead of directly sequencing the liver of control and fasted groups. Since fasting significantly induced the mRNA expression of IGFBP2 in the liver compared to the normally-fed control, it would be beneficial to explain the reasoning behind this choice.
Response: Thank you very much for your questions. We performed IGFBP2 overexpression in goose primary hepatocytes as we would like to identify the genes and pathways that are regulated by IGFBP2. Transcriptome sequencing of the livers from the fasting/refeeding geese can identify a batch of genes and pathways that are affected by fasting or refeeding. In this study, however, it is more important to identify the genes and pathways that are regulated by IGFBP2 other than the genes and pathways that are affected by fasting or refeeding. Identification of the genes and pathways regulated by IGFBP2 could help address the function of IGFBP2 in liver cells and the role of IGFBP2 in the response of liver to fasting/refeeding, so we did not perform the transcriptome sequencing analysis on the livers from the fasting/refeeding geese in this study. I hope the reviewer agree to this explanation.
Comment 3: In Figure 9, the authors concluded that IGFBP2 affects growth and inflammation, but there is no production data mentioned in the full text. Therefore, the authors should consider whether this is an appropriate graph.
Response: Thanks for your comments. We have deleted the statement of ‘growth and inflammation’ from the graph.
Comment 4: It is unclear how to explain the difference in the effects of overexpression of IGFBP2 (Fig.4) and siIGFBP2 (Fig.5B) on the Cytokine-Cytokine Receptor Pathway. Although the results suggest that siIGFBP2 has a stronger effect on certain gene promotion (LOC106041919 and CCL20), overexpression has a more substantial influence than siIGFBP2 on gene expression (Fig.3A) than siIGFBP2 (Fig.5A).
Response: Good point! My interpretation to the difference is shown as follows: Firstly, IGFBP2 can exert its role in IGF-dependent or independent manner. When IGFBP2 functions in IGF-dependent manner, the effects of overexpressed IGFBP2 will be limited by the amount of IGF. In contrast, reduced expression of IGFBP2 by siRNA is not the case, i.e., the effects of reduced IGFBP2 is not restricted by the amount of IGF in culture medium. Secondly. The protein level of IGFBP2 in the cells transfected with IGFBP2 overexpression vector is restricted by the output of translation machinery. In contrast, it is not the case for IGFBP2 knocking down. As the functions of IGFBP2 are dependent on the level of IGFBP2 protein, the effects of IGFBP2 on certain gene expression (LOC106041919 and CCL20) are also restricted by the efficiency of translation machinery. Thirdly, IGFBP2 is a secretary protein, so its effects are regulated by secretion process. In summary, the limitations on the effects of IGFBP2 most likely take place in the cells transfected by IGFBP2 overexpression vectors but not in the cells transfected with siIGFBP2, which may cause a stronger effect on the expression of LOC106041919 and CCL20 in the IGFBP2-knocking down cells than in IGFBP2-overexpressing cells. In addition, the off-target effects of siRNA cannot be ruled out. I hope the reviewer agree to this interpretation.
Comment 5: The clarity of Figure 3B is insufficient. Please improve the quality of the image.
Response: Thanks. Done as requested.
Comment 6: References 2, 20, 25, 38, 39, 45, 51, and 52 have formatting issues such as incomplete page numbers or missing volume numbers, and they should be corrected according to American English standards.
Response: Thanks for your comments. We have revised it accordingly. Due to changes in the content, the original References 2, 20, 25, 38, 39, 45, 51, and 52 have been respectively adjusted to References 2, 15, 20, 19, deleted, 38, 44, and 45.
Reviewer 2 Report
The study presented in the abstract explores the impact of changes in nutritional and energy status on hepatic IGFBP2 expression and its role as a mediator. The findings indicate that overfeeding inhibits IGFBP2 expression in the liver, while fasting induces its expression. Additionally, the study demonstrates that insulin can inhibit IGFBP2 expression in goose primary hepatocytes, suggesting a potential link between blood insulin levels and IGFBP2 expression in the liver. Transcriptome sequencing analysis further reveals that the overexpression of IGFBP2 in geese primary hepatocytes leads to significant changes in the expression of multiple genes, particularly those associated with cytokine-cytokine receptor, immune, and lipid metabolism-related pathways. These findings are validated by in vivo geese models and IGFBP2-siRNA treatment in goose primary hepatocytes. Overall, this study provides valuable insights into the role of IGFBP2 in mediating the biological effects induced by variations in nutritional and energy levels through the cytokine-cytokine receptor pathway.
1 Please provide additional information regarding the primer sequences in Table 1. Line 282 does not contain the required details about the primer sequences.
2 Please include all differentially expressed genes as an attachment for lines 403 to 404. In addition, Table 1 does not provide the corresponding genes. Furthermore, it is necessary to provide the gene name, gene symbol, log-fold change, p-value, and adjusted p-value for each gene.
3 Please provide the necessary information, including the data volume and mapping rate for RNAseq
4 In the "2.6 Transcriptome Analysis" section of this study, please provide the specific version of the goose genome that was used.
5 Please provide the reference gene(s) used as internal controls in the Materials and Methods section. Additionally, indicate the specific reference gene(s) utilized in Figure 1, Figure 2, Figure 3A, Figure 3C, and any other relevant figures.
6 Line 321, "softwar" should be "software".
7 The title does not reflect the entire content of the research. Please modify the title and main text according to the content.
Some editing for English language is required throughout the manuscript due to same mistakes
Author Response
Comments and Suggestions for Authors
The study presented in the abstract explores the impact of changes in nutritional and energy status on hepatic IGFBP2 expression and its role as a mediator. The findings indicate that overfeeding inhibits IGFBP2 expression in the liver, while fasting induces its expression. Additionally, the study demonstrates that insulin can inhibit IGFBP2 expression in goose primary hepatocytes, suggesting a potential link between blood insulin levels and IGFBP2 expression in the liver. Transcriptome sequencing analysis further reveals that the overexpression of IGFBP2 in geese primary hepatocytes leads to significant changes in the expression of multiple genes, particularly those associated with cytokine-cytokine receptor, immune, and lipid metabolism-related pathways. These findings are validated by in vivo geese models and IGFBP2-siRNA treatment in goose primary hepatocytes. Overall, this study provides valuable insights into the role of IGFBP2 in mediating the biological effects induced by variations in nutritional and energy levels through the cytokine-cytokine receptor pathway.
Response: Thank you very much for your positive comments and constructive suggestions, which can help us to improve this manuscript significantly. We have carefully read and consider these critiques and suggestions, and made revisions to the manuscript accordingly. The point by point responses are shown as follows:
Comment 1: Please provide additional information regarding the primer sequences in Table 1. Line 282 does not contain the required details about the primer sequences.
Response: I am very sorry for my carelessness. The information on the primer sequences is actually in the Supplementary Table 1 other than Table 1. We missed the word “Supplementary” in the old version of manuscript.
Comment 2: Please include all differentially expressed genes as an attachment for lines 403 to 404. In addition, Table 1 does not provide the corresponding genes. Furthermore, it is necessary to provide the gene name, gene symbol, log-fold change, p-value, and adjusted p-value for each gene.
Response: Thanks for your suggestion. Actually, all the relevant data from transcriptome sequencing analysis have been submitted to the GenBank databases, please see the Data Availability Statement in the manuscript. Anyway, we have included the information as an attachment for your reference in this submission(please see Attachment 1). In regard to Table 1, I am very sorry for my carelessness again. Table 2 should be Table 1. I have corrected the mistake. Please see Line 342. For other information, such as the gene name, gene symbol, log-fold change, p-value, and adjusted p-value for each gene, has been provided as requested.
Comment 3: Please provide the necessary information, including the data volume and mapping rate for RNAseq.
Response: Done as requested. Please see Line 336-338 and Supplementary Table 2.
Comment 4: In the "2.6 Transcriptome Analysis" section of this study, please provide the specific version of the goose genome that was used.
Response: Done as requested. Please see in Line 280.
Comment 5: Please provide the reference gene(s) used as internal controls in the Materials and Methods section. Additionally, indicate the specific reference gene(s) utilized in Figure 1, Figure 2, Figure 3A, Figure 3C, and any other relevant figures.
Response: Thanks for your suggestion. We have provided the internal control gene in the Materials and Methods section (Line 252) and in the notes of the figures as you indicated (please see Line 317, 329, 368, 391, 416, 429, 444 and 468).
Comment 6: Line 321, "softwar" should be "software".
Response: Thank you very much for your meticulousness. Done as requested.
Comment 7: The title does not reflect the entire content of the research. Please modify the title and main text according to the content.
Response: Thanks for your comments. We have revised the title as “Goose hepatic IGFBP2 is regulated by nutritional status and participates in energy metabolism mainly through the cytokine-cytokine receptor pathway”, and the main text is also revised accordingly.
Round 2
Reviewer 2 Report
Dear authors
I am pleased to say that I am highly satisfied with the responses to my queries and concerns.
The meticulous attention to detail and the comprehensive revisions you made have significantly strengthened the manuscript. the responses adequately addressed the points raised in my previous review, and you have effectively clarified certain aspects that were previously ambiguous. I commend the dedication to improving the quality of the research.
Based on the revised manuscript and the satisfactory responses, I would like to recommend its acceptance for publication in Animals. The content is now well-structured, the arguments are well-supported, and the findings are valuable to the scientific community. I believe that this manuscript will make a significant contribution to the field and generate fruitful discussions among researchers.
Best wishes